# Vocational Interventions to Improve Employment Participation of People with Psychosocial Disability, Autism and/or Intellectual Disability: A Systematic Review

**DOI:** 10.3390/ijerph182212083

**Published:** 2021-11-17

**Authors:** Isabelle Weld-Blundell, Marissa Shields, Alexandra Devine, Helen Dickinson, Anne Kavanagh, Claudia Marck

**Affiliations:** 1Melbourne School of Population and Global Health, University of Melbourne, Melbourne 3010, Australia; isabelle.weldblundell@unimelb.edu.au (I.W.-B.); marissa.shields@unimelb.edu.au (M.S.); a.kavanagh@unimelb.edu.au (A.K.); claudia.marck@unimelb.edu.au (C.M.); 2School of Business, University of New South Wales, Canberra 2610, Australia; h.dickinson@adfa.edu.au

**Keywords:** systematic review, randomized control trials, vocational interventions, psychosocial disability, autism, intellectual disability, employment

## Abstract

**Objective:** To systematically review interventions aimed at improving employment participation of people with psychosocial disability, autism, and intellectual disability. **Methods:** We searched MEDLINE, Embase, PsycINFO, Web of Science, Scopus, CINAHL, ERIC, and ERC for studies published from 2010 to July 2020. Randomized controlled trials (RCTs) of interventions aimed at increasing participation in open/competitive or non-competitive employment were eligible for inclusion. We included studies with adults with psychosocial disability autism and/or intellectual disability. Risk of bias was assessed using the Cochrane Collaboration Risk of Bias II Tool. Data were qualitatively synthesized. Our review was registered with PROSPERO (CRD42020219192). **Results:** We included 26 RCTs: 23 targeted people with psychosocial disabilities (*n* = 2465), 3 included people with autism (*n* = 214), and none included people with intellectual disability. Risk of bias was high in 8 studies, moderate for 18, and low for none. There was evidence for a beneficial effect of Individual Placement and Support compared to control conditions in 10/11 studies. Among young adults with autism, there was some evidence for the benefit of Project SEARCH and ASD supports on open employment. **Discussion:** Gaps in the availability of high-quality evidence remain, undermining comparability and investment decisions in vocational interventions. Future studies should focus on improving quality and consistent measurement, especially for interventions targeting people with autism and/or intellectual disability.

## 1. Introduction

The centrality of paid employment in supporting people with and without disabilities to meet socio-economic needs, maintain health and well-being, and engage in civil and political participation is widely recognized [1,2,3]. Indeed, there is evidence to suggest the benefits of employment for people with disabilities are more significant than for people without disabilities [4,5]. Similarly, the negative effects of unemployment (socio-economic disadvantage poorer health) may be greater for people with disabilities due to existing socio-economic disparities [6,7,8,9]. Despite the socio-economic and health rationale, and commitments made by governments through treaties such as the United Nations Convention on the Rights of Persons with Disabilities, gaps in employment rates between those with and without disabilities persist in many countries [10,11,12].

In the US, for example, 29% of people with disabilities aged 16–64 years are employed, compared to 70% of those without disabilities [13], with these patterns replicated in the United Kingdom [14] and Australia [15].While people with disabilities as a whole tend to have poorer labor force outcomes than the population without disabilities, individuals with certain types of disabilities, such as psychosocial disability (i.e., disability that may arise from mental health conditions), autism (i.e., development condition significantly affecting communication and social interaction), and/or intellectual disability (i.e., difficulties with intellectual functioning (learning, problem solving) and adaptive functioning (communication, independent living)) may fare particularly poorly [16]. In the UK, 33% of people with psychosocial disability, 22% of people with autism, and 27% of people with a severe or specific learning disability (noting data was not disaggregated by intellectual disability more broadly) are reported to be employed [17], compared to 52.3% of people with disabilities more broadly [14]. These employment rates are similar to those seen in Australia, where only 25.7% of people with psychosocial disability are employed and 7.9% are unemployed [18], 38% of people with autism are in the labor force with 34.1% unemployed [19], and 32% of individuals with intellectual disability employed and 6.9% unemployed [16].

Numerous vocational interventions (e.g., developing job capacity, identifying suitable work) to improve employment outcomes for people with disability have been developed. One of the most well-document vocational interventions is Individual Placement and Support (IPS)/Supported Employment (SE) [3,20,21,22,23,24]. IPS/SE was designed and widely practiced in the US to support people with Severe Mental Illness (SMI) and is increasingly implemented in other OECD contexts. There have also been small scale trials of IPS with other cohorts, such as returned veterans, adults within the justice system, and young people with autism [25,26,27,28,29].

Less has been written about interventions for people with autism and/or intellectual disability. What has been documented generally focuses on tailoring individualized job opportunities such as through Customized Employment (CE) [30,31,32]. CE is described as a person-centered approach comprising two key stages: (1) Discovery, whereby the skills, aspirations and employment opportunities of an individual are explored; and (2) Job Carving, whereby individuals/practitioners engage employers to identify and negotiate suitable open/competitive employment opportunities that meets the priorities and on-the job support needs of employers and employees [30,31,32].

Closing the gaps in employment outcomes requires evidence on what types of vocational interventions are effective for which groups of people with disabilities. Given those with psychosocial disability, autism, and/or intellectual disability often experience poorer employment outcomes, determining evidence of what works for these cohorts is of utmost importance [15,33]. Therefore, this paper systematically reviews RCTs of vocational interventions that aimed to improve employment outcomes of people with psychosocial disability, autism, and/or intellectual disability, to inform understanding about what intervention investments may work best for these cohorts.

## 2. Materials and Methods

This systematic review was registered with PROSPERO (CRD42020219192). This review followed the PRISMA 2020 checklist for reporting systematic reviews (Appendix A).

### 2.1. Eligibility Criteria

For inclusion, studies had to include participants with a medical diagnosis of autism or intellectual disability, and/or meet the following definition of psychosocial disability: a medical diagnosis of ≥1 mental illness that impedes participation in employment, and at least 75% of study participants with psychosocial disabilities must have a severe mental illness (SMI). In accordance with the Royal Australian and New Zealand College of Psychiatrists, we considered SMI as schizophrenia and other psychoses, bipolar disorder, severe depression, and severe anxiety [34]. We excluded studies that included individuals with other diagnoses that did not separately report data on participants with psychosocial disability, autism, and/or intellectual disability. Consistent with other reviews in the area [20,35,36], we excluded studies whose definition of psychosocial disability was substance use disorder without any other mental illness. To be eligible for inclusion, at least 75% of study participants had to be of working age, between 16 and 64 years [37]. At the start of the intervention, participants could be not in work for any time-length or employed but looking for additional work.

We included RCTs of interventions with a vocational component aimed at increasing participation in open/competitive or supported/non-competitive employment and reported on this primary outcome of interest at follow-up. We were deliberately inclusive in relation to eligible interventions, which ranged from skills-development, career counselling, work experience, work placement programs, active labor market programs, and workplace practices. We included studies where the comparison group either received no intervention (i.e., passive or wait list control) or received an active control intervention that met our definition of an eligible intervention. Studies of interventions with a medical component (e.g., use of antidepressants) were excluded, unless that component was given to individuals in both intervention and control arms. We included international evidence from high-income countries, as defined by the World Bank [38]. We only considered research published in English and in peer-reviewed journals. In the event of insufficient RCT data, we planned to search non-randomized studies of vocational interventions. Further information on this additional search and analysis is detailed in Appendix A.

### 2.2. Information Sources, Search Strategy and Selection Process

Eight databases (MEDLINE, Embase, PsycINFO, Web of Science, Scopus, CINAHL, ERIC, ERC) were searched for studies published in English from 2010 to 5 July 2020 inclusive (see Appendix A for search strategy). The reference lists of four relevant systematic reviews were also searched [24,36,39,40]. After the removal of duplicates, three researchers (IWB, MS and CM) independently reviewed titles and abstracts first, and then full text articles using Covidence, so that each article was screened by two researchers. Disagreement was resolved through discussion [41].

### 2.3. Data Extraction and Data Items

Data was extracted by one reviewer (IWB) into a pre-piloted Excel spreadsheet template. A second reviewer (MS) independently extracted data for randomly selected studies for comparison of interrater reliability. Discrepancies were resolved through discussion with a third reviewer (CM). Data were extracted on study design and methods, study sample, intervention and control setting, primary outcomes, secondary outcomes (if applicable), method of statistical analysis, and key results. The primary outcome of interest was open/competitive, or non-competitive/job in supported setting at follow-up closest to the end of the intervention but no longer than 12 months following the end of the intervention. Where available, we assessed information on secondary outcomes of interest, including sustained employment (participants who have maintained employment at a follow-up point closest to but longer than 12 months post-intervention), job satisfaction using a validated tool among people with disabilities, and work readiness at pre- and post-intervention using a validated tool among people with disabilities. When required data were not reported, we contacted study authors for further information. When multiple analyses were reported, we extracted the more conservative analysis, with a focus on intention-to-treat analyses.

### 2.4. Risk of Bias Assessment

Risk of bias was assessed using the Cochrane Collaboration Risk of Bias II (RoB-II) Tool [42]. The RoB-II uses a series of signalling questions to assess bias arising from five domains: (1) randomization process; (2) deviations from intended interventions; (3) missing outcome data; (4) measurement of the outcome; and (5) selection of the reported result. For each domain, the authors determined if the study is at ‘low risk of bias’, ‘some concerns’, or ‘high risk of bias.’ The authors then determined the overall risk of bias, which corresponds to the highest risk of bias in any of the included domains. One reviewer (IWB) evaluated the RoB of included studies. A second researcher (MS) independently assessed the RoB for a random selection of 10% of included studies to ensure interrater reliability. Discrepancies were resolved through discussion.

### 2.5. Synthesis

Key results on the primary and secondary outcomes of this review were described by disability type in summary tables and through narrative synthesis. An initial goal of our review was to pool the available evidence and conduct a meta-analysis, but we were unable to do so as a result of the heterogeneity in the conceptualization and measurement of the outcomes.

## 3. Results

### 3.1. Study Characteristics

The results of searching, screening, and full-text review are shown in Figure 1. After full-text review, 29 articles were identified for inclusion in the review, relating to 26 unique RCTs, as three articles reported on sustained employment outcomes from original RCTs [43,44,45]. The excluded articles and reasons for exclusion are shown in Appendix A. There were 23 studies of participants with psychosocial disabilities (*n* = 2465) and three studies of participants with autism (*n* = 214). No RCTs pertaining to people with intellectual disability were identified. Additional searches for non-randomized interventions for people with autism or intellectual disability resulted in only two studies insufficient for synthesis (see Appendix A).

The characteristics of the RCT studies included can be seen in Table 1. Most studies with people with psychosocial disabilities included individuals with a variety of mood, anxiety, and/or psychotic disorders [27,29,46,47,48,49,50,51,52,53,54,55,56,57,58]. Three studies included participants with schizophrenia only [59,60,61], three studies exclusively included individuals with psychosis [62,63,64], and in one study all participants had post-traumatic stress disorder (PTSD) [28]. In the three studies including participants with autism, youth were assessed as having autism with a medical diagnosis of ASD or an educational identification of autism [45,65,66]. Nearly half of the studies included were performed in the USA (*n* = 12) [27,28,29,45,49,53,55,59,60,61,65,66]. Three studies were carried out in the UK [54,63,67], three in Canada [48,52,64], two in Japan [51,58], two in Australia [57,62], and one in each of Sweden [46], Switzerland [47], the Netherlands [50], and Hong Kong [56].

IPS was the most common intervention in studies among people with psychosocial disabilities, featuring in 15 of 23 studies [27,28,43,46,50,51,52,54,56,57,60,61,62,63,64]. In the eight remaining studies, interventions included: Job Coach based on a modified IPS model [44]; IPS plus skills-development [49]; work-related skills-development [59]; and career guidance [53]. The only interventions assessed for people with autism was Project SEARCH and ASD Supports [65,66,68]. The duration of interventions ranged widely, from 5–10 business days [29,59] to 5 years [44], with some studies placing no limits on intervention duration [27,50]. In two studies the intervention duration was unclear [51,54]. A detailed description of the vocational interventions and comparison conditions in included studies is provided in Appendix A.

Twenty-five of the studies included reported the proportion of participants in open employment, although employment was operationalized with varying definitions and lengths of follow-up. The only study not to report open employment instead provided the paid employment rate from the end of intervention to 30 days post-intervention [29]. For further information on the interventions and outcome definitions, see Table 2.

### 3.2. Risk of Bias Assessment

Figure 2 shows the domain-level risk of bias judgments for included studies with people with psychosocial disabilities and autism; Figure 3 presents the risk of bias assessments across each domain. A total of 15 studies among people with psychosocial disabilities were assessed to have ‘some concerns’ [27,28,29,43,44,46,49,50,51,52,56,58,60,62,64]; eight were at high risk of bias [48,53,54,55,57,59,61,63]. All three studies including participants with autism had ‘some concerns’ [65,66,68]. None of the 26 studies included were at low risk of bias. Common reasons for ‘some concerns’ were lack of published study protocols outlining outcome definitions and measurement and/or statistical analysis plans. Issues with missing outcome data and potential selection of outcome definition or reported results were common issues leading to a high risk of bias assessment.

### 3.3. Psychosocial Disability

#### 3.3.1. Primary Outcomes: Studies with Moderate Risk of Bias

Of the 15 studies with moderate risk of bias, six examined the effect of IPS on open employment compared to traditional vocational rehabilitation (TVR) [28,43,46,50,51,62], and all found that a greater proportion of individuals in the IPS intervention reported open employment. However, the magnitude of the effect and definition of open employment varied across studies. Defining open employment success as working for at least one week, Bejerholm et al. found that 46.3% of IPS participants worked compared to 10.9% of TVR recipients [46]. Among veterans, Davis et al. found that 76.2% of IPS participants reported at least one day of work during a 12-month period, compared to 27.9% of TVR participants [28]. In Heslin et al.’s study, 22.1% of IPS participants were continuously employed for at least 30 days, compared to 11.6% of TVR recipients [43]. Considering open employment as working for a minimum of one day in the previous six-month period, Killackey et al. found that 71.2% of IPS participants worked, compared to 48.0% of TVR recipients [62]. Using a similar definition of open employment, 43.7% of IPS participants worked compared to 25.2% of TVR recipients in Michon et al.’s study [50]. Finally, Oshima et al. found that a greater proportion (44.4%) of IPS participants obtained open employment, defined as five or more work hours per week, than individuals in the TVR condition (10.5%) [51].

A total of five studies compared IPS to conditions other than TVR, with most observing improved outcomes for those in the IPS group. Bond et al. found more participants in the IPS condition (31.0%) worked in open employment, defined as working at least one day, compared to those in the control condition who received a job club-style program (7.0%) [27]. At six-month follow up, Nuechterlein et al. did not find a difference in the proportion of participants in open employment, defined with no minimum number of days or hours, between those receiving IPS and vocational rehabilitation plus social skills training [60]. However, at 7 to 18 months post-intervention a greater proportion of individuals in the IPS group worked (69%) compared to those in the control group (33%). Two studies compared IPS to a control condition wherein participants could use other employment services or seek employment by any means of their choice. Erickson et al. did not observe a difference in the proportion of individuals in open employment for at least one day between the two groups [64], while Poremski et al. found that more participants in IPS obtained employment (34%) compared to those in the control condition (22%) [52]. Tsang et al. compared traditional IPS to IPS plus work-related social skills training (Integrated Supported Employment, ISE), as well as to TVR [56]. Results showed that at 15-month follow-up, 74.1% of participants in ISE worked in open employment (continuously worked in a job for >=2 months for at least 20 h per week), compared to 44.6% in IPS and 6.1% in TVR.

Two studies examined job coach interventions. Davis et al. compared the effect of standard coaches compared to vocational coaches on the rate of individuals in paid employment over a thirty day period, observing no difference in paid employment between the two groups (42.9% vocational coaches, 28.6% standard coaches) [29]. Hoffman et al. similarly assessed a Job Coach intervention compared to TVR, and found that participants in the Job Coach intervention were more likely to work in open employment for at least two weeks over a five year period (65.2% v 33.3%) [44].

A further two studies examined Supported Employment programs in conjunction with additional interventions. McGurk et al. assessed the effect of enhanced Supported Employment plus the Thinking Skills for Work Program compared to enhanced Supported Employment only. A greater proportion of individuals in the intervention were in open employment from 0 to 24 months post-baseline (60%) compared to those in the control group (36%) [49]. In a study comparing the effect of cognitive remediation and Supported Employment to TVR, Yamaguchi et al. found that most participants in the intervention were in work for at least 1 day in the 12 month follow-up period (62.2%), compared to those in the control condition (19.1%) [58].

#### 3.3.2. Primary Outcomes: Studies with High Risk of Bias

Of the eight studies with high risk of bias, four examined the effect of IPS or IPS plus on employment outcomes. In Waghorn et al.’s study comparing IPS to TVR, at 12-month follow-up more participants in the IPS condition (42.5%) obtained employment, defined as attending at least one day of paid work, than control participants (23.5%) [57]. Similarly, Twamley et al., 2012 found that more IPS participants were in any paid employment during the 12-month study (56.7%) compared to individuals receiving TVR (28.5%) [61]. Schneider et al., 2016 compared IPS plus (including a work-focussed counselling intervention) to IPS only, with competitive employment defined as working for at least one hour when data was collected at 12 months post-baseline. The authors found no difference in employment between the intervention (41%) and control group (29%) [54]. Craig et al., 2014 also compared IPS plus (including motivational interviewing) to IPS only. Despite the conceptualization of employment being unclear, the authors found that more participants in the intervention condition were in open employment from baseline to 12 months (43%) compared to individuals in the control condition (18%), and on the day of interview at 12 months more participants in the intervention group were in open employment (38.3%) compared to those in the control group (15.2%) [63].

Two studies examined the effect of virtual reality job interview training (VR-JIT) compared to TAU waitlist control on accepted job offers at 6 months post-intervention. Smith et al. (2015a) found that more participants in the VR-JIT group accepted job offers compared to control participants (38.5% v 25.0%) [55]. These results were similar to those reported in Smith 2015b, wherein 39.1% of participants in the intervention group accepted job offers compared with 14.3% of control participants [59]. One study assessed the effect of a vocational empowerment photovoice intervention compared to wait-list control on open employment at 4.5 months post-intervention and found no difference in outcomes between participants in the intervention (14%) and control (4%) group [53].

Lecomte et al.’s (2020) study considered the effect of a cognitive behaviour therapy group intervention adapted for supported employment program plus supported employment (CBT-SE) compared to supported employment only and found that a greater proportions of individuals in the CBT-SE intervention had been in open employment (defined as a minimum of one week) (75.0%) compared to individuals in the control group (57.8%) at 12-month follow-up [48].

#### 3.3.3. Secondary Outcomes: Moderate Risk of Bias

Only two studies, Davis et al. [28] and Hoffman et al. [44], clearly reported on sustainment of employment outcomes (i.e., defined by authors as working in competitive employment/at least 50% of work-time in competitive employment for at least a number of weeks) with IPS participants more likely to sustain employment in a competitive/open job compared to those in TVR. Two studies examined self-esteem, both using the Rosenberg Self Esteem questionnaire. Heslin et al. found no evidence of differences in self-esteem between the IPS and TVR groups at either 12- or 24-months follow-up [43]. Michon et al. likewise found no significant differences in the self-esteem of participants in the IPS and TVR groups at 30 months follow-up [50].

A further two studies assessed program costs. Both Hoffman et al. and Yamaguchi et al. found no differences in total costs between the intervention and control groups [44,58]. However, Hoffman et al. noted that participants in the Job Coach intervention had significantly higher incomes than the control group, while Yamaguchi found that the mean costs for medical services in the cognitive remediation and supported employment intervention group were lower. None of the studies including people with autism examined any of the secondary outcomes of interest. None of the included studies including people with psychosocial disabilities or autism formally assessed barriers and/or facilitators for implementation of interventions.

#### 3.3.4. Secondary Outcomes: High Risk of Bias

Two of the high risk of bias studies included information on secondary outcomes of interest. Evidence from Russinova et al.’s study using a photovoice intervention found that individuals who received the intervention had a significantly greater increase in overall empowerment as measured by The Empowerment Scale, including self-efficacy, compared to controls [53]. Schneider et al., 2016 assessed cost-effectiveness and found no significant differences between the IPS plus and IPS only groups. This study also assessed self-esteem using the Rosenberg Self Esteem questionnaire and found no differences in self-esteem between the two groups at 12-month follow-up [54].

### 3.4. Autism

#### Primary Outcomes: Moderate Risk of Bias

All three studies including people with autism were of moderate risk of bias and found participants in the Project SEARCH plus ASD supports group had improved employment outcomes compared to participants receiving the control condition, high school special education services as usual. In the Wehman 2017 and 2020 studies, intervention participants were more likely to be in open employment, although the proportions varied across the studies (74.2% and 31.6% respectively, compared to 5.6% and 4.8% amongst controls) [65,68]. The outcome was conceptualized differently in Whittenburg et al., wherein the authors found that more intervention recipients accepted job offers for open employment (83.3%) than did participants in the control group (12.5%) [66]. None of the studies with people with autism measured secondary outcomes of interest to this review.

## 4. Discussion

Our systematic review of the effectiveness of vocational interventions identified 23 RCTs targeting people with psychosocial disabilities, three RCTs for people with autism, and no RCTs that focused on people with intellectual disability. A further search for non-randomized intervention studies for people with autism or intellectual disability only identified two studies, not considered sufficient for synthesis.

Similar to other reviews [3,20,22,24], regardless of the potential risk of bias, studies found evidence for a beneficial effect of IPS for people with psychosocial disabilities compared to TVR or other control conditions on open employment outcomes in almost all relevant studies (11/12), although beneficial effects were not always observed over longer periods [43,62]. Aligning with earlier reviews, there was also some evidence for IPS plus interventions over IPS only or another intervention [20]. In addition, there was evidence from several single studies that interventions using a vocational program component increased open employment outcomes [28,43,46,50,51,57,61,62], but these need replication to increase confidence in the results.

For people with autism, there is some evidence that intensive job training (including profiling, skills training, long-term individualized support) alongside placement programs (job development, internships and on-the-job training) delivered in the final year of secondary school resulted in open employment [31,45,65,66,68]. This evidence was from three studies with young people aged 18–21 years, pointing to a lack of intervention studies in people with autism older than 21 years. Trials assessing the effectiveness of vocational interventions for people with intellectual disability are still completely lacking [39,40].

Overall, our study reveals concerns in relation to the heterogeneity of interventions studied, and quality of evidence reviewed. Alongside missing outcome data and potential for selective reporting, all studies included were assessed as having a moderate or high risk of bias. A central concern was few of the included RCTs previously published a priori study protocols with clearly defined outcomes or how they were to be measured or analyzed. If a priori study protocols were more readily available, this would reduce potential outcome measure selection and selective reporting and would increase transparency and confidence in the reliability of the results. Further, the variation in how employment was defined and measured, and at what timepoint post-intervention it was measured across studies, makes it difficult to assess or make comparisons as to the effectiveness of different interventions. For example, primary outcome definitions of ‘employment success’ ranged from participants receiving a job offer within 0–6 months [59], to paid open employment for at least 30 continuous days over a one- and two-year period [43,67]. These discrepancies make it difficult to compare the effectiveness of even similar interventions. Similar variation was observed in the length of intervention reported, which ranged from not reported [51], 5–10 business days [55], to five years [44,47]. Future research should therefore focus on establishing consistent standards of outcome measurements, and publishing study protocols a priori, including outcome definition, measurement, and statistical analysis plans.

Only two studies included in our review reported on any cost-effectiveness measures, neither of which reported significant differences between the intervention and control groups [44,58]. Yet, in one of these studies, participants in the Job Coach intervention group had higher incomes compared with the TVR group [44], while in the Yamaguchi study lower medical services costs for the intervention were reported, alongside a high probability for cost-effectiveness in terms of vocational outcomes [58]. We note that, more broadly in the literature, there have been a small number of studies demonstrating the cost-effectiveness of IPS and/or IPS supplemented with cognitive remediation and social skills training, in relation to both vocational outcomes and health service utility outcomes [69,70,71,72].

Compared to workers without disabilities, those with disabilities are more likely to report poorer psychosocial working conditions (e.g., job satisfaction, job control) [4,73]. Not only does this have a negative impact on mental health, but it reduces the likelihood that people will remain in employment [1,6,73]. This highlights the importance of getting people with disabilities into decent work (i.e., secure, fair conditions, enables social and personal growth) and measuring outcomes associated with employment quality [74]. Secondary outcomes of interest to this review (e.g., job satisfaction, self-esteem), however, were not commonly reported. Inconsistent reporting on whether employment was sustained is of significant concern when assessing whether interventions help people into jobs that can be maintained. There was some consistency in how studies measured self-esteem. For example, three studies used the Rosenberg Self Esteem questionnaire [43,54,67,75]. Albeit no differences were reported between the impact of the intervention and control arms on self-esteem across these studies: raising questions as to whether these interventions are superior to TVR in supporting people with disabilities to find and maintain decent work that improves socio-economic and mental health outcomes. Again, these issues point to the urgent need to develop meaningful outcome definitions and measures in this field, that can be applied consistently to research and real-world outcomes.

Concurring with our review findings, most of the available evidence on the effectiveness of vocational interventions is focused on trial-based IPS for people with SMI [76]. Non-trial based research that has been conducted indicates that IPS holds its effectiveness outside of trials and may be effective for other groups; with the strongest evidence directed towards veterans with PTSD, and pilot program data of a small cohort of young people with ASD supportive of further trialing and research [3,25,26,76,77]. Overwhelmingly, however, these studies agree there is a need for more rigorous and longer-term evaluation of how IPS may be applied effectively beyond its current focus. There are also calls for research as to whether vocational interventions can be effectively implemented within emerging systems such as through the UK Personalised Budget programs and the Australian National Disability Insurance Scheme (NDIS) [21,78].

The NDIS provides individualized budgets to eligible Australians with disability to purchase services and supports to achieve self-determined goals. A key premise of the NDIS is that it will lead to improved employment outcomes as participants can use their budgets to access supports to build their capacity to find and maintain employment [79]. This has resulted in an emerging market of vocational interventions that have the potential to provide more individualized employment approaches, including through Customised Employment (CE). To date, there is limited research as to whether vocational interventions can be effectively implemented within emerging systems such as the NDIS or UK equivalent of personalized budgets for people with disability [21,78]. Mapping the availability and key components of these emerging approaches and evaluating their effectiveness is therefore critical to addressing the evidence gap, particularly in our understanding of what works to help people with autism and/or intellectual disability to find and maintain employment [80].

As acknowledged in the limitations of our review, the individualized and multifaceted nature of vocational interventions targeted towards people with autism and/or intellectual disability, may make it more difficult to conduct RCTs on their effectiveness. One approach to addressing this gap is to identify successful interventions and closely analyze the key reasons for their success through impact and outcome evaluations, particularly where formative, process, and output evaluations have already indicated potential effectiveness [81]. Other evaluation techniques that have been applied to interventions such as CE include quasi-experimental design that compare outcomes for individuals who have received CE, with a comparison group of similarly aged people with comparable disability types identified in larger data sets [82]. Qualitative research of experiences and outcomes of vocational intervention participants and those who deliver them, will also deepen our understanding. It remains imperative that consistent outcome definitions and measures are applied within emerging intervention programming and evaluation frameworks, so that comparative analysis of the effectiveness of different interventions is possible.

While not included in the analysis, our secondary search of non-randomized vocational interventions identified two studies that highlight key components for supporting people with autism and/or intellectual disability [83,84]. For example, Langi et al. found specialized transition programs that integrated school and community-based training within work places, that were delivered during and after secondary school, were more effective than TVR in helping young people disability, including those with intellectual disability, obtain employment outcomes [83]. Similar to the RCTs for young people with autism included in our systematic review analysis, key components of effective programs included job-readiness development, job-shadowing, on-the-job training and support, funded work experience, and job coaching [83]. In addition, Kaya et al. found people with autism who received vocational interventions were more likely to find employment when compared to those who did not receive these supports, with increasing number of supports received, leading to greater odds of finding employment [84].

Barriers to employment are often multifaceted and cumulative, intertwining individual socio-demographic characteristics (e.g., type, severity and episodic nature of disability), and vocational (e.g., limited skills, work experience) and non-vocational barriers (e.g., poor mental health, homelessness, financial difficulties, availability of family or community support), with situational (e.g., limited confidence of employers, insufficient on-the-job supports and workplace accommodations) and contextual and structural barriers (e.g., limited supply of jobs, stigma and discrimination) to gaining and maintaining work [85,86]. Understanding of how different individual-level barriers and contextual factors may influence the effectiveness of vocational interventions is paramount [86,87]. While many studies described barriers to work for people with disabilities, only a few studies have formally assessed and reported on individual-level work barriers or factors influencing intervention effectiveness. The most commonly-reported individual-level barriers to work included failure to engage with or disengagement from vocational interventions [27,43,51,67], poor mental health, and co-occurring health conditions [27,29,48,60]. Intervention implementation effectiveness was influenced by the level of collaboration, with better outcomes achieved when employment specialists were well-supported by other stakeholders such as schools and employers. The resource-intensive nature of interventions such as Project SEARCH, alongside the need for highly skilled employment specialists, similarly challenged implementation [45]. The difficulty of integrating mental health and employment services was specifically highlighted as undermining IPS implementation in the Australian context [57].

More broadly, studies emphasized structural barriers such as flat labor markets (as has been widely observed during the ongoing COVID-19 pandemic) and less protective socio-economic contexts. Less generous disability benefits have also been shown to influence the effectiveness of IPS and other vocational interventions across many OECD contexts [43,50,67]. The considerable length of time, and large range, taken to obtain employment even within intervention arms suggests that the effectiveness of interventions is substantially weakened by the interplay of individual-level and contextual barriers experienced by job seekers with disability. This underscores not only the need for investment in the most effective interventions, but wider policy reform which addresses multi-faceted vocational, non-vocational and structural barriers [76,87,88] to achieve improved employment participation for all people with disability.

## 5. Conclusions

The pre-determined inclusion/exclusion criteria applied to our systematic review and subsequent quality assessment enabled a critical analysis of the status of the evidence base on vocational interventions, demonstrating that even well-established studies could be strengthened to promote a priori reporting of study design and consistency of measurement. In accordance with systematic review guidelines, the stringent criteria applied meant that RCTs beyond the scope of this review were not considered, even though they may have been useful to discussion. We also focused on studies from high-income countries only, meaning that our findings are less applicable across non-high-income settings. The variability in target groups, operationalization of interventions, and definition and measurement of primary outcomes, combined with the general low-moderate quality of studies, proved incompatible to meta-analysis or GRADE assessment. Gaps in the availability of high-quality evidence remain, undermining comparability and future investment decision-making, particularly for people with autism and intellectual disability. For people with psychosocial disability, we found evidence for a beneficial effect of IPS, IPS plus other interventions, and some supported employment interventions on open employment outcomes. For people with autism, there is some evidence of benefit for Project SEARCH and ASD Supports on open employment participation, arising from studies with people aged 18–21 years only.

Experiences and severity of health conditions, impairment, and/or mental illness and any associated disability are extraordinarily heterogenous in nature. While not a specific objective of this systematic review, we do highlight that while many studies provided baseline data disaggregated by differential diagnoses of mental illness and/or level of severity, employment outcome measures were not similarly disaggregated. This makes it more difficult to examine the effectiveness of vocational interventions for people with different types and severities of conditions and/or disability. This remains a gap in the evidence base worthy of further research.

Regarding our secondary outcomes of interest, few studies assessed sustained employment, work readiness, job satisfaction, and cost-effectiveness, and no studies formally assessed barriers and facilitators for implementation. All studies included were assessed as having a moderate or high risk of bias. Future efforts should be focused on establishing consistent standards of outcome measurement for employment interventions in this area, and publishing study protocols including outcome definition, measurement, and statistical analysis plans. Where the comparatively limited scale, availability, and multifaceted nature of vocational interventions targeted towards people with autism and/or intellectual disability make it difficult (but not impossible with sufficient will and appropriate resources) to conduct RCTs, investing in mixed method impact and outcome evaluations that incorporate consistent and comparable measures is essential to enhancing the evidence base on what works to support all people with disability gain and maintain employment.

## Figures and Tables

**Figure 1 ijerph-18-12083-f001:**
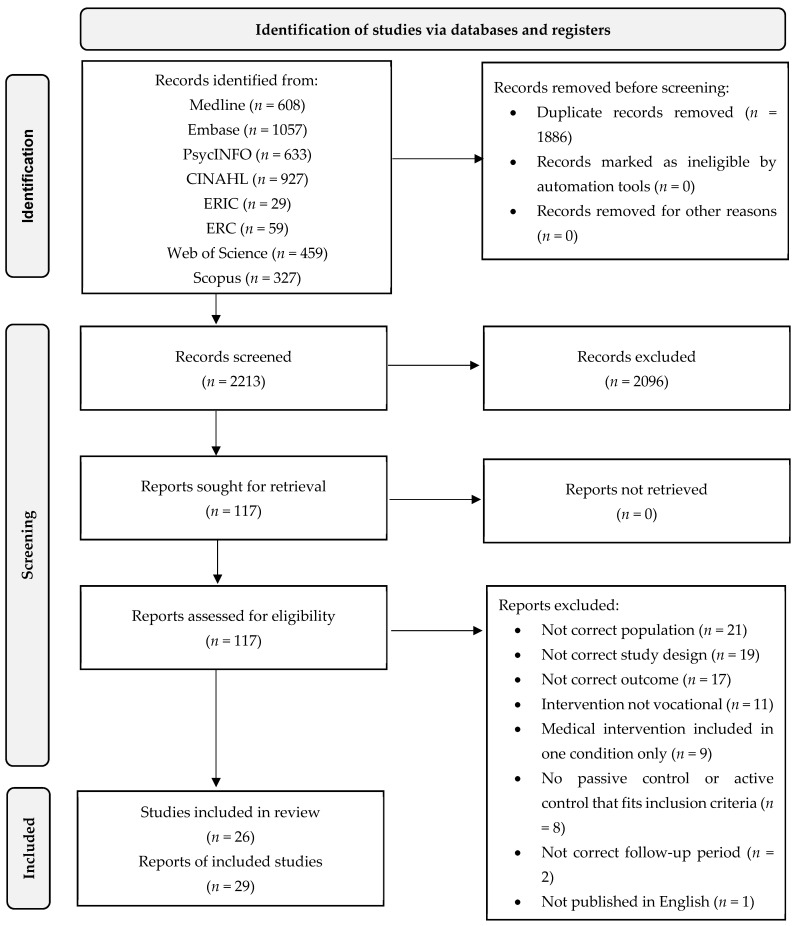
PRISMA flow diagram.

**Figure 2 ijerph-18-12083-f002:**
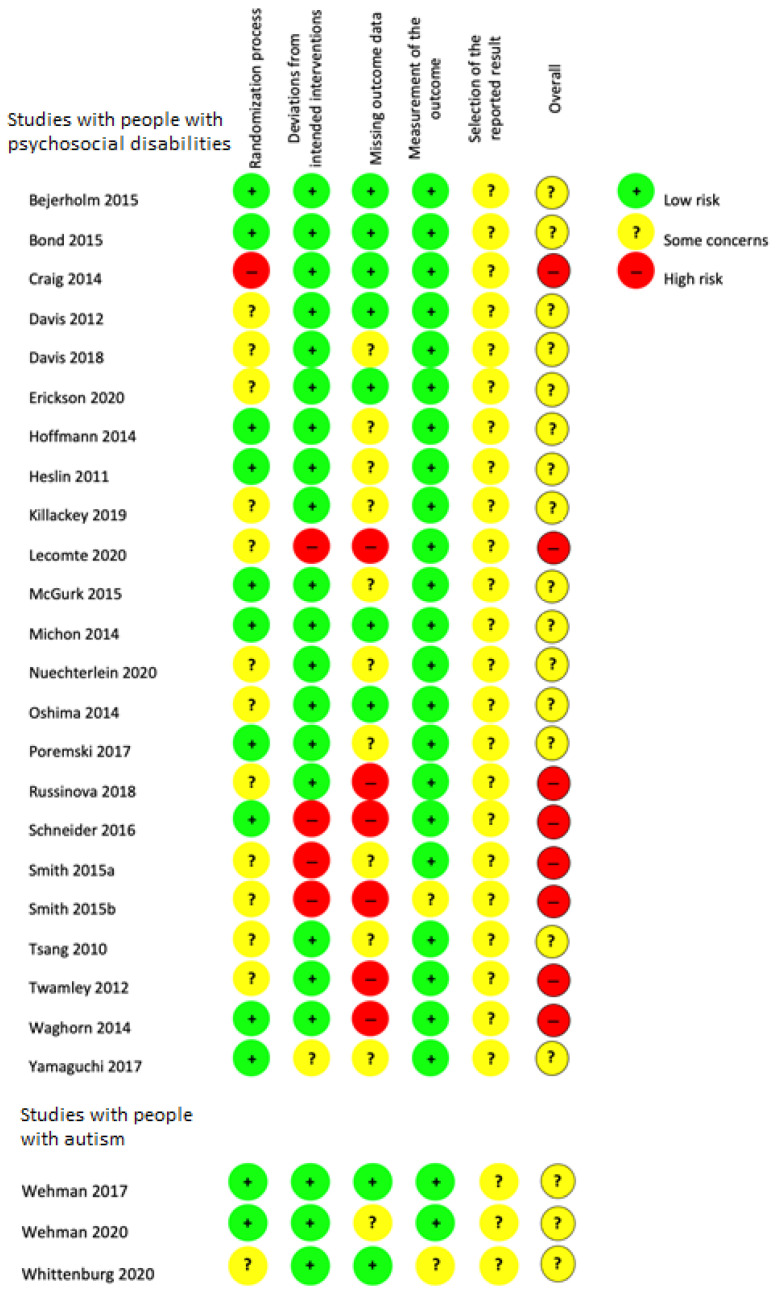
Risk of bias in included studies.

**Figure 3 ijerph-18-12083-f003:**
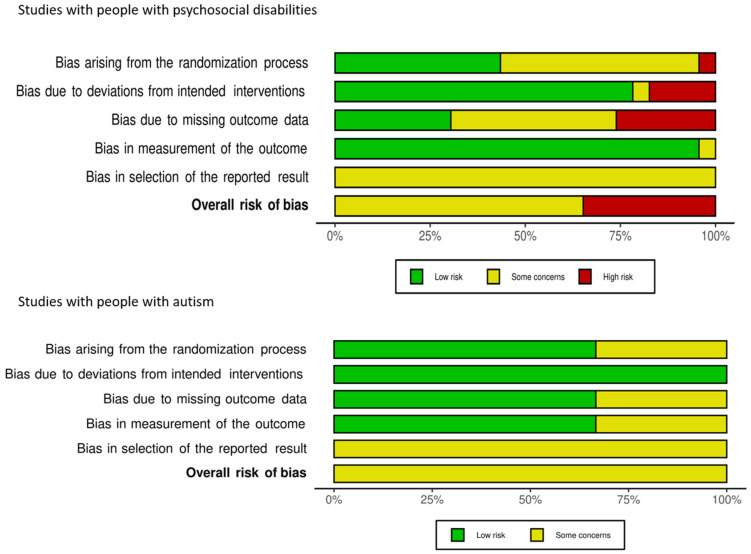
Risk of bias across each bias domain.

**Table 1 ijerph-18-12083-t001:** Characteristics of included studies.

Studies with People with Psychosocial Disabilities						
Study, Country	N Total (Control/Intervention)	Male Participants (%)	Age Range	Mean Age (SD) Total, Control, Intervention	Description of Psychosocial Disabilities (%)	Employment Status at Baseline
Bejerholm 2015, Sweden [46]	120 (60/60)	55.8	18–63	Total: Not reported (NR) C: 38 (8) I: 38 (8) (only provided whole digits)	64.2% schizophrenia and other psychosis (ICD-10 F20–29), 7.5% bipolar (ICD-10 F31), 27.5% other diagnoses (ICD-10 F32 F40 F60)	Had not worked in the preceding year
Bond 2015, USA [27]	90 (45/45) *n* = 3 dropped post-randomization and sample reduced to 87 (1 control, 2 intervention)	79.3	18 or older	Total: NR C: 44.6 (11.6) I: 42.9 (11.5)	53% schizophrenia, 18% depressive disorder, 25% bipolar disorder, 3% other (information not available)	No competitive employment in past three months
Craig 2014, UK [63]	159 (78/81)	73.0	18–35	Total: NR C: Midlands 2: 24 (3.7); London 2: 24 (4.7) I: Midlands 1: 23 (4.2); London 1: 25 (4.2) (only provided whole digits)	100% early psychosis	Unemployed
Davis 2012, USA [28]	85 (43/42)	88.2	19 to 60	Total: NR C: 40.5 (12.5) I: 39.9 (11.9)	100% post-traumatic stress disorder, 89% major depressive disorder, 20% dysthymia, 54% agoraphobia, 59% panic disorder, 28% social phobia, 42% alcohol dependence, 21% alcohol abuse, 37% drug dependence, and 18% drug abuse	Unemployed
Davis 2018, USA [29]	32 (16/16)	78.1	17–20	Total: 17.8 (NR) C: 17.9 (NR) I: 17.6 (NR)	50% major depressive disorder, 25% anxiety disorder, 16% bipolar disorder, 9% state MH Authority Services	Employed or unemployed
Erickson 2020, Canada [64]	109 (53/56)	82.6	18–30	Total: NR C: 22.7 (3.3) I: 23.4 (3.5)	4.6% schizophreniform, 37.6% schizophrenia, 8.3% schizo-affective disorder, 18.4% bipolar, 9.2% major depression, 15.6% Psychosis NOS, 4.6% substance-induced psychosis, 0.9% delusional disorder, 0.9% Aspergers	Unemployed or employed and seeking better jobs
Hoffmann 2012, Switzerland [47]	100 (54/46)	65.0	18–64	Total: NR C: 34.1 (9.2) I: 33.5 (9.8)	38% schizophrenia spectrum, 41% affective disorder, 21% other, 12% concomitant substance abuse	Out of competitive employment
Howard 2010, UK [67]	219 (110/109)	67.1	18–65	Total: NR C: 38.3 (9.3) I: 38.4 (9.5)	72.5% psychotic disorder; 27.5% mood disorder	Unemployed for at least 3 months
Killackey 2019, UK [62]	146 (73/73)	69.2	15–25	Total: 20.4 (2.4) C: 20.5 (2.1) I: 20.4 (2.7)	100% psychotic disorder including 43.8% schizophreniform/schizophrenia, 13.0% schizoaffective disorder, 11.6% major depressive disorder with psychotic features, 13.7% bipolar disorder, 11.6% psychosis not otherwise specified, 6.2% other	Unemployed or employed
Lecomte 2020, Canada [48]	164 (85/79) Data recorded as treated, not as intention to treat.	60.7%	NR	Total: 36.6 (11.3) C: 37.0 (11.6) I: 36.1 (11.0)	Severe mental illness (schizophrenia, bipolar, or major depression). Primary diagnoses: 18.5% Dx mood disorder, 7.4% Dx anxiety disorder, 0.6% Dx organic disorder, 58.6% Dx psychotic disorder, 1.2% Dx substance-related, 6.2% Dx personality disorder, 1.9% Dx developmental disorder, 5.6% Dx other.	Currently not working and seeking work, or working less than 5 h a week and wishing for another job with more hours
McGurk 2015, USA [49]	107 (50/57)	75.4	NR	Total: 44.1 (11.0) C: 42.9 (10.7) I: 45.1 (11.3)	23.4% schizophrenia, 22.4% schizoaffective disorder, 23.4% bipolar disorder, 16.8% major depression, 14.0% other	Not worked in past 3 months, or exited competitive job that lasted <3 months
Michon 2014, Netherlands [50]	151 (80/71)	74.2	18–65	Total: NR C: 35.6 (11.0) I: 34.1 (9.9)	Clients of long-term mental health care and at baseline 93% of participants were diagnosed with one or more specific mental disorders. 50.6% psychotic disorder. Remaining participants had various diagnoses, such as enduring major depression, personality disorders, developmental disorders. At baseline 7% was assessed by mental health care professionals as ‘diagnosis postponed’ or ‘no diagnosis available’	No paid work
Nuechterlein 2020, USA [60]	69 (23/46)	66.7	18–45	Total: 24.5 C: 25.1 (3.8) I: 24.2 (4.2)	84% schizophrenia, 14% schizoaffective disorder, depressed type, mainly schizophrenic, 2% with schizoaffective disorder, manic type, mainly schizophrenic	Employed or unemployed
Oshima 2014, Japan [51]	37 (19/18)	75.7	18–59	Total: NR C: 41.1 (9.4) I: 40.1 (8.5)	Primary diagnosis of either schizophrenia, mood disorder, or neurotic disorder	Not competitively employed
Poremski 2017, Canada [52]	90 (45/45)	63.3	18 or older	Total: NR C: 47.1 (10.6) I: 45.2 (9.4)	64% major depressive disorder, 22% psychotic disorder, 6% panic disorder, 4% mania-hypomania, 3% post-traumatic stress disorder	Not working
Russinova 2018, USA [53]	55 (29/26 N = 4 (C:2, I:2) excluded from analysis as already in receipt of employment services)	39.2	18 or older	Total: NR C: 45.3 (14.2) I: 47.0 (10.9)	31.4% schizophrenia/schizoaffective, 31.4% bipolar, 33.3% bipolar, 2.0% post-traumatic stress disorder and anxiety/panic disorder, 2.0% personality disorder, 2.0% post-traumatic stress disorder, anxiety/panic disorder and personality disorder	Not working
Schneider 2016, UK [54]	74 (37/37)	70.3	18–60	Total: NR C: 29.5 (NR) I: 30.5 (NR)	43.2% psychosis, 23.0% schizophrenia, 14.9% bipolar disorder, 13.5% depression, 4.1% other	Not currently in work
Smith 2015a, USA [55]	25 (8/17) N = 32 with participants from previous RCTs included in analysis (11 control, 21 intervention)	53.1	18–55	Total: NR C: 39.1 (10.6) I: 40.8 (12.2)	100% schizophrenia or schizoaffective disorder	Unemployed or underemployed
Smith 2015b, USA [59]	70 (22/48)	NR (68.6 at 6 months)	18–65	Total: NR C: 49.1 (10.9) I: 47 (12.4)	45.1% posttraumatic stress disorder, 47.1% major depressive disorder, 33.3% bipolar disorder, 15.7% schizophrenia or schizoaffective disorder	Unemployed or underemployed
Tsang 2010, Hong Kong [56]	189 (IPS 65, Integrated Supported Employment 58, TVR 66)	49.2	NR	Total: NR C: 36.5 (7.6) I: 34.1 (9.0)	76.7% schizophrenia, 23.3% other	Unemployed
Twamley 2012, USA [61]	58 (28/30)	63.8	45 or older	Total: 51 (SD NR) C: 51.8 (5.1) I: 50.3 (3.5)	40% schizophrenia, 60% schizoaffective disorder	Unemployed
Waghorn 2014, Australia [57]	208 (102/106)	69.2	18–59	Total: NR C: 32.8 (8.9) I: 32.0 (8.9)	80.8% psychotic disorder, 8.2% bipolar affective disorder, 6.3% major depression or anxiety disorder	Not employed within the previous three months
Yamaguchi 2017, Japan [58]	111 (54/57)	62.0	20–45	Total: 35 (SD NR) C: 34.5 (6.8) I: 34.8 (7.1)	87.0% schizophrenia, 7.6% major depression or 5.4% bipolar disorder	Unemployed
**Studies with People with Autism**						
**Study, Country**	**N Total (Control/Intervention)**	**Male Participants (%)**	**Age Range**	**Mean Age (SD) Total, Control, Intervention**	**Type of Disability (%)**	**Employment Status at Baseline**
Wehman 2014, USA [45]	44 (20/24) 4 assigned to control group dropped out prior to study, so C = 16.	72.5	18–21	Total: NR C: 19.1 (1.1) I: 20.0 (1.1)	Autism (ASD diagnosis and/or educational eligibility of Autism)	Unemployed
Wehman 2020, USA [65]	156 (75/81)	76.0	18–21	Total: NR C: 19.5 (1.2) I: 19.8 (1.1)	Autism	Unemployed
Whittenburg 2020, USA [66]	14 (8/6)	78.6	18–21	Total: NR C: NR I: NR	Autism. Participants with comorbid intellectual disability and/or mental health disorders were included	Unemployed

Notes: Not reported (NR).

**Table 2 ijerph-18-12083-t002:** Interventions and outcomes of included studies.

Studies with People with Psychosocial Disabilities								
Study	Interventions	Intervention Categories	Duration of Intervention, Months	Follow-Up, Months after Randomisation (Unless Otherwise Stated)	Definition of Primary Outcome and Measurement (Timepoint/Period in Months)	Primary Outcomes	Secondary Outcomes	Results
Bejerholm 2015, Sweden [46]	IPS vs. Traditional vocational rehabilitation	SE vs. Skills development	18	18	Open employment defined as worked for at least 1 week in employment that paid at least minimum wage, available to any citizen and located in mainstream settings (0–18)	More IPS participants worked than participants in the TVR group (19/41, 46.3% versus 5/46, 10.9%, respectively; Difference (95% CI): 36 (18–54); *p* < 0.001)		Favors IPS
Bond 2015, USA [27]	IPS vs. Work Choice	SE vs. Career guidance	No fixed duration	12	Open employment (Worked at least one day in the community for which an individual is paid at least minimum wage during 0–12) Supported employment, agency-run job (worked at least one day, an agency-run, for profit business that sells products or goods to the public and provides supported employment to disabled individuals during 0–12) or sheltered work (at least one day, transitional and/or long-term employment in a controlled and protected working environment for those who are unable either to compete or to function in the open job market due to their disabilities during 0–12 months postintervention)	More participants in the IPS condition worked in open employment than those in the control condition (13/42, 31.0% versus 3/43, 7.0%; N = 85, χ2 = 7.99, df = 1, *p* < 0.01) More participants in the control than IPS condition were in sheltered employment (1/43, 2.3% versus 0/42, 0% respectively)		Favors IPS for open employment, no effect for sheltered employment
Craig 2014, UK [63]	IPS and motivational interviewing vs. IPS only	SE and career guidance vs. SE	12	12	Open employment (12, 0–12 months)	More participants in the IPS and motivational interviewing condition were in open employment from baseline to 12 months than participants in the IPS only condition (29/68, 43% vs. 12/66, 18% respectively; OR = 3.5, 95% CI 1.5–8.1). and on the day of interview at 12 months (26/68, 38.2% vs. 10/66, 15.2%, respectively; χ2 = 8.79, df = 1, *p* = 0.003)		Favor IPS and MI over IPS only
Davis 2012, USA [28]	IPS (fair fidelity) vs. Veteran Affairs Vocational Rehabilitation Program	SE vs. Career guidance and work experience	12	12	Open employment (job for regular wages in a setting that was not set aside, sheltered, or enclaved. Day labor (that is, pick-up cash-based day jobs for yard work, babysitting, manual labor, and so forth) and military drill were not counted as competitive employment, at least one day (any number of hours) of actual work during (0–12 months)	More participants assigned to IPS obtained open employment compared to the TVR participants (76.2% vs. 27.9%, number needed to treat = 2.07, 95% CI = 1.96–2.19; χ2 = 19.84, df = 1, *p* < 0.001)	IPS participants worked higher mean number of weeks in a competitive job compared to control group (21.6 (17.7) vs. 6.8 (13.8), *p* < 0.001 (Mann-Whitney z), Cohen’s d = 0.93, 95% CI = 0.50–1.36)	Favors IPS
Davis 2018, USA [29]	Standard Coaches vs. Vocational Coaches	Skills development vs. Skill development	6–16 (depending on client needs)	1 month post-intervention, 4 months post-intervention	Paid employment in the 30-day period from the end of intervention to 1-month post-intervention	There was no difference in paid employment between participants with vocational coaches compared with standard coaches (6/14, 42.9% versus 4/14, 28.6%, *p* = 0.430)		No effect
Erickson 2020, Canada [64]	IPS vs. No constraints on the use of other employment support services	SE vs. Passive control condition	12	6, 12	Open employment, at least 1 day of work (0–6, 6–12)	No effect from baseline to 6 months follow-up between the IPS and control conditions (30/50, 60% vs. 30/52, 57.7%, respectively; no effect from 6 months to end of intervention at 12 months follow-up (34/48, 72.3% versus 25/51, 50.0%) (please note that the *n* and % don’t add up, and authors did not clarify)		No effect
Hoffmann 2012, 2014, Switzerland [44,47]	Job Coach vs. Traditional train-place vocational rehabilitation programs	SE vs. Work experience and skills development	60	24, 60	In open employment for at least 2 weeks over the 5-year study (60, 0–60) Supported employment (24)	Participants in the Job Coach condition were more likely to work in open employment than TVR participants over the 5-year study period (30/46, 65.2% vs. 18/54, 33.3%) (*p* = 0.002); and on the day of interview at 5 years follow-up (20/46, 43.5% vs. 9/54, 16.7%; *p* = 0.002). Participants in the control condition were more likely to be in sheltered employment than those in the supported employment condition (control 19/54, 35% vs. SE 12/46, 26%; Sign. < 0.001)	Intervention participants were more often employed at least 50% (130 weeks) in a competitive job (SE 20/46, 43.5% vs. control 6/54, 11.1%, respectively; *p* < 0.001). There were no significant differences in vocational program or mental health service costs between the groups. However, participants in the Job Coach condition had significantly higher income than controls	Favors Job Coach for open employment, but not sheltered employment
Howard 2010, UK [67] Heslin 2011, UK [43]	IPS vs. Local traditional vocational services	Supported employment (SE) vs. skills development and career guidance	24	24	Open employment defined as a job paying at least the minimum wage, located in a mainstream socially integrated setting not set aside for persons with disabilities, held independently (i.e., was not agency owned) and the participant was in continuous employment for at least 30 days (with parttime employment rated pro-rata) (0–12, 0–24)	More participants in the IPS condition were in open employment from baseline to 24 months, compared to participants in the TVR condition (IPS 22.1% vs. TVR 11.6%, risk ratio 1.91; 95% CI 0.98 to 3.74; *p* = 0.053; adjusted analysis *p* = 0.041)	There were no differences in self-esteem as measured with the Rosenberg Self Esteem questionnaire at 12 months (*p* = 0.90) or 24 months *p* = 0.47) There was no difference in job satisfaction at the 12 month follow-up using the Indiana Job Satisfaction Scale between employed participants in the two groups (*p* = 0.29) Cost-effectiveness showed no substantial differences.	Favors IPS
Killackey 2019, UK [62]	IPS (Good fidelity) vs. Referral to external government-contracted employment agencies	IPS vs. TVR	6	6	Open employment defined as working in a job in the open labor marker that paid the legislated minimum wage for a minimum of 1 day in the previous 6-month period (0–6).	IPS participants were more likely to work compared to control participants (47/66, 71.2% versus 29/60, 48.0% respectively; OR = 3.40, 95% CI 1.17–9.91, z = 2.25, *p* = 0.025)		Favors IPS
Lecomte 2020, Canada [48]	Cognitive behaviour therapy group intervention adapted for supported employment programs (CBT-SE) plus supported employment program vs. supported employment program only.	SE and skills development vs. SE	1 month	12	Open employment (0–12). A minimum of one week.	Participants in the CBT-SE intervention were more likely to work than those in the SE only condition (57/76, 75.0% versus 37/64, 57.8% respectively; *p* < 0.05; OR 2.2, 95% CI: 1.0, 4.8). please note, this was not an intention-to-treat analysis, but rather -as treated.		Favors CBT-SE
McGurk 2015, USA [49]	Enhanced supported employment plus the Thinking Skills for Work Program vs. Enhanced supported employment only.		6	24	Open employment (0–24 months) Any paid employment (0–24).	More intervention participants than control participants were in open employment from 0 to 24 months from baseline (34 out of 57, 60% vs. 18 out of 50, 36%) *p* = 0.02; More IPS participants than control participants were in paid employment from 0 to 24 months from baseline (37 out of 57, 65% vs. 22 out of 50, 44%) from 0 to 24 months from baseline (*p* = 0.03)		Favors Thinking Skills for Work Program
Michon 2014, Netherlands [50]	IPS (moderate to good) vs. Traditional vocational rehabilitation	Skills development and work experience	No limit (although a limit of 36 months is often prescribed by financing systems in the Netherlands)	30 (last follow-up timepoint)	Open employment (worked in a competitive job for one day or more) (competitive employment was defined as having a paid job in a company or organization in the regular labor market, against prevailing wages, not set aside for persons with a disability, that is, in an integrated work setting). (0–30 months)	More IPS participants worked compared to TVR participants, (31/71, 43.7% versus 20/79, 25.3%, *p* < 0.05.)	The Rosenberg Self Esteem questionnaire showed no significant difference between IPS and traditional vocational rehabilitation at 30-month follow-up	Favors IPS
Nuechterlein 2020, USA [60]	IPS plus Workplace Fundamentals Module vs. Conventional Brokered Vocational Rehabilitation plus social skills training intervention		18	6, 18	Open employment defined as paid work in a job that was open to applications from the general public (competitive employment), no minimum number of days of employment, but typically participants were employed at least several weeks.” (1–6, 7–18 months)	There was no difference in open employment between participants in the intervention condition compared to control condition in the initial 6-month period (7/22, 32% versus 12/41, 29%, respectively), From 7 months to end of intervention at 18 months more intervention participants worked compared to controls (69% vs. 33%, respectively, (Adjusted analysis, *p* = 0.02)		No effect for initial 6 months, Favors IPS + WFM –for the following 1-year period
Oshima 2014, Japan [51]	Good IPS vs. Conventional vocational rehabilitation	Skills development and work experience	NR	6	Open employment defined as a job paying at least minimum wage (as established in Japanese law), with five and more work hours per week, for which anyone can apply, and not controlled by a service agency. (0–6) Supported employment (0–6)	Participants in the IPS condition were more likely to obtain open employment than those in the control group (44.4% versus 10.5% respectively; *p* = 0.022) There was no difference in supported employment rates between IPS participants and TVR participants (2/18, 11.1% versus 0/19, 0%, respectively; *p* = 0.128)		Favors IPS for open employment, no effect for supported employment
Poremski 2017, Canada [52]	IPS vs. free to seek employment by any means of their choice		Entire intervention: 27 Good fidelity: 8	27	Open employment (20–27, during the 8 months of good fidelity IPS)	More participants in the IPS condition obtained employment than those in control condition (34% vs. 22%, respectively; *p* = 0.16; adjusted analysis showed that participants in the IPS group had a 2.4 (*p* = 0.02) greater chance of obtaining employment		Favors IPS
Russinova 2018, USA [53]	Vocational Empowerment Photovoice (high fidelity) vs. Wait-list control		Approx. 4.5 months	4.5 (postintervention), 7.5 (3 months post-intervention)	Open employment: having at least one day on the job (point prevalence of competitive employment at postintervention and 3 months postintervention)	There was no difference in open employment between participants in intervention condition and waitlist controls at postintervention (14% vs. 4% respectively; Cohen’s d = 0.75)	Participants in the intervention condition had a s greater increase in overall empowerment compared to waitlist controls, including self-efficacy (overall empowerment: Group effects: F = 6.65, *p* = 0.01e, Effect size (Cohen’s d) = 0.39) (self-efficacy subscale: Group effects: F = 6.08, *p* = 0.02, Cohen’s d= 0.30)	No effect
Schneider 2016, UK [54]	IPS plus work-focused counselling intervention vs. IPS only.		12	12	Open employment (0–12)	There was no difference in employment between groups (intervention 41% vs. control 29%; χ2 = 0.73, *p* = 0.39)	No difference in the Rosenberg Self Esteem questionnaire at 12-month follow-up. There was no strong evidence for cost-effectiveness	No effect
Smith 2015a, USA [55]	Virtual reality job interview training (VR-JIT) vs. TAU waitlist control	Skills development	5–10 business days	6 months postintervention	Open employment (accepted job offers during 0–6 months postintervention)	More participants in the virtual reality group accepted job offers compared to control participants. (38.5% vs. 25.0%, no statistical analysis)	None	Potentially favors VR-JIT (no statistical analysis)
Smith 2015b, USA [59]	Virtual reality job interview training vs. Waitlist control		2 weeks	6 months postintervention	Open employment (accepted job offers during 0–6 months postintervention)	More participants in the virtual reality job interview training group accepted job offers compared with control participants. (39.1% vs. 14.3%, no statistical analysis)	None	Potentially favors VR-JIT (no statistical analysis)
Tsang 2010, Hong Kong, [56]	Integrated Supported Employment (ISE): IPS and work-related social skills training vs. IPS vs. Traditional vocational rehabilitation (TVR) (good fidelity)	SE and skills development vs. SE vs. career guidance and skills development/work experience	TVR: 15 ISE, IPS: 39	15, 39	Open employment (competitive employment, continuously worked in the job for > = 2 months for at least 20 h per week) (0–15, 0–39)	There were significant differences between the three groups at 15 month follow up (end of TVR) (TVR 4/66 6.1% vs. IPS 29/65 44.6% vs. Integrated Supported Employment 43/58 74.1%, *p* < 0.001, More participants in the ISE condition worked compared to the IPS condition at 39 months (ISE 48/58 82.8% vs. IPS 40/65 61.5%, *p* = 0.009)		Favors ISE over IPS at both timepoints and IPS and ISE over TVR at the first timepoint
Twamley 2012, USA [61]	IPS (Fair to good fidelity) vs. Conventional vocational rehabilitation	SE vs. career guidance and skills development	12	12	Open employment defined as employment paying at least minimum wage and not reserved for the disabled. “We only considered someone employed if they worked for any part of a day.” (0–12 months); Any paid employment (0–12)	More IPS participants were in open employment than those in TVR during the 12-month study (56.7% versus 28.6% respectively; *p* = 0.031)More IPS participants obtained paid employment than TVR participants (70.0% versus 35.7%, respectively; *p* = 0.009)		Favors IPS
Waghorn 2014, Australia [57]	IPS vs. Non-integrated forms of supported employment		12	12	Open employment (0–12)	More participants in the IPS condition obtained open employment than control participants (42.5% versus 23.5% respectively; OR (95% CI) = 2.40 (1.32, 4.36), *p* < 0.01).		Favors IPS
Yamaguchi 2017, Japan [58]	Cognitive remediation and supported employment vs. traditional vocational services	Skills development + SE vs. TVR	NR, waiting for author’s response	12	Open employment: number of people who worked at least 1 day in competitive work at 12 months follow-up	More participants in the cognitive remediation and supported employment condition were in work compared to those in the traditional vocational services condition (62.2% versus 19.1% respectively), *p* < 0.001; adj OR = 11.06 (95% CI 3.53–34.62)	There was no difference in mean total costs between the groups, however, the mean cost for medical services in the intervention group was lower. Further, the intervention showed high probability for cost-effectiveness in terms of vocational outcomes.	Favors CR + SE
**Studies with People with Autism**								
**Study**	**Interventions**	**Intervention Categories**	**Duration of Intervention, Months**	**Follow-Up, Months after Randomisation**	**Definition of Outcome and Measurement (Timepoint/Period in Months)**	**Primary Outcomes**	**Secondary Outcomes**	**Results**
Wehman 2014, 2017, USA [45,68]	Project SEARCH plus ASD Supports vs. High school special education services as usual	Collaborative, employer-based employment training and placement program	9	9 (post-intervention)	Open employment (9)	Intervention participants were more likely to be in competitive employment than control participants at graduation (74.2%, 23/31 versus 5.6%, 1/18, respectively; *p* < 0.0001)		Favors SE + ASD
Wehman 2020, USA [65]	Project SEARCH plus ASD Supports vs. High school special education services as usual	Collaborative, employer-based employment training and placement program	9	9	Open employment (9)	Intervention participants were more likely to be in employment than control participants, (31.6% vs. 4.8% *p* < 0.001; adj RR 5.84, 95% CI 1.50, 13.3, *p* = 0.014)		Favors SE + ASD
Whittenburg 2020, USA [66]	Project SEARCH plus ASD Supports vs. High school special education services as usual	Collaborative, employer-based employment training and placement program	NR	12	Accepted job offers for open employment (0–12) Sheltered work (12)	More intervention participants accepted job offers compared to control group participants (83.3% vs. 12.5%, no statistical analyses, very small group). At the 12-month follow-up, one of the control participants and none of the PS + ASD participants was in sheltered work		Probably favors SE + ASD for open employment (no statistical analysis performed), no effect for sheltered employment.

Note: Supported Employment (SE); Individual Placement and Support (IPS), Traditional Vocational Rehabilitation (TVR).

## Data Availability

All data presented in this review are available in the included papers.

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
