# Peer review of "Vocational Interventions to Improve Employment Participation of People with Psychosocial Disability, Autism and/or Intellectual Disability: A Systematic Review"

_ijerph, 2021, doi:10.3390/ijerph182212083_

Round 1

Reviewer 1 Report

Isabelle Weld-Blundell, Marissa Shields, Alexandra Devine, Helen Dickinson, Anne Kavanagh, and Claudia Marck summarized in this review, the status and effectiveness of the vocational interventions designed to improve employment participation of people with psychosocial disability, autism and/or intellectual disability, underlining their potential benefits in improving employment outcomes. Given the increased prevalence of ASD, the increasing population of young adults with ASD is a challenging public health issue. There is a great disparity between the estimated costs associated with ASD, including unemployment, and health care utilization costs, and the evidence base for understanding what interventions can optimize employment and other important functional outcomes for individuals with ASD can be of great importance. The article gives an interesting scientific perspective on the gaps in the availability of high-quality evidence, especially for interventions targeting people with autism and/or intellectual disability, underlining the difficulties to compare and make investment decisions in vocational interventions in these conditions, so the paper is extremely relevant for the field. It addresses many issues and brings a summary of the existing data. The review followed the PRISMA guideline, the study was registered in the international prospective register of systematic reviews (PROSPERO) and the results were properly presented.

I was wondering if the authors could develop a bit more the discussion and conclusion focusing on the issues that could make employment challenging for people with diverse severe and persistent mental illnesses. The rationale for that would be the fact that, people with severe mental illnesses find maintaining employment challenging for a range of complex reasons, such as stigma and discrimination, lack of support in the workplace, or issues with fluctuating health conditions, access to proper care, family, and financial support.  Some of the problems are general, but some of the difficulties may be specific for each disorder and developing a vocational intervention program efficient for all these disorders included under the term “psychosocial disability”, is unlikely. Context is also critical, so describing study contexts, including employment markets, support structures, and welfare policies may play an important role in outcomes.

I recommend the article for publication.

Author Response

Thank you so much for your review of our paper and for the insightful recommendations. Please find attache our response to the review. Sincere thanks, Alexandra Devine

Reviewer 2 Report

Thank you for the opportunity to review this paper.

Title: Vocational interventions to improve employment participation of people with psychosocial disability, autism and/or intellectual disability: a systematic review.

General recommendations:

  • It is a relevant and applicable study that provides indications for the implementation programmes to improve employment participation of people with psychosocial disabilities.
  • It is suitable for publication, but I would like to make some suggestions.

TITTLE

Because autism can be considered a psychosocial disability and because no papers were found referring to people with intellectual disability, the title could be shortened. For example:

  • Vocational interventions to improve employment participation of people with psychosocial disability: a systematic review.

INTRODUCTION SECTION

Perhaps autism, intellectual disability and psychosocial disability definitions should be included in the introduction, explaining their specific characteristics and difficulties. The concept of psychosocial disability is especially diffuse and needs to be delimited.

RESULTS SECTION

3.1. Study characteristics

The results of searching, screening and full-text review are shown in Figure 1.

Figure 1. PRISMA flow diagram. Authors are advised to consider using the 2020 PRISMA proposal, as there are some differences with respect to the previous version.: http://prisma-statement.org/prismastatement/flowdiagram.aspx

  • In the author column of table 1 and 2, the name of the author and the year should also be included. For example: Bejerholm 2015, Sweden [46].
  • It is suggested to asterisk the reference list in all those papers included in the current review. For example:

* 46. Bejerholm, U., et al., Individual placement and support in Sweden—a randomized controlled trial. Nordic journal of psychiatry, 2015. 613 69(1): p. 57-66.

3.2. Risk of Bias assessment

In the risk of bias section, in addition to Figure 2, another figure could be included with the risk of bias summary plot. For example:

  • https://bookdown.org/MathiasHarrer/Doing_Meta_Analysis_in_R/risk-of-bias-plots.html

Conclusion section

As a reflection, it can be noted that:

  • To get information on barriers and facilitators for the implementation of intervention programs, qualitative studies should be consulted: in educational contexts there is little tradition of randomized controlled trials.

Regards

Author Response

Thank you very much for your review of our paper and the very useful recommendations. Please see our attached response to the reviewers. In sincere thanks, Alexandra Devine
